# Mechanical Properties and Microstructures of Laser–TIG Welded ME21 Rare Earth Mg Alloy

**DOI:** 10.3390/ma12132188

**Published:** 2019-07-08

**Authors:** Taotao Li, Gang Song, Zhaodong Zhang, Liming Liu

**Affiliations:** School of Material Science and Engineering, Dalian University of Technology, Key Laboratory of Liaoning Advanced Welding and Joining Technology, Dalian 116024, China

**Keywords:** ME21 Mg alloy, AZ31 Mg alloy, columnar crystals, microstructure, microhardness

## Abstract

The microstructural and mechanical properties of laser–tungsten inert gas (TIG) hybrid welding of Mg alloy sheets for automobiles are investigated in the present work, including AZ31 and ME21, AZ31 and AZ31, ME21 and ME21, and corresponding comparisons were carried out. The results show that columnar crystals appear in the ME21/ME21 and ME21/AZ31 heat-affected zones, and no columnar crystals formed in the AZ31/AZ31 fusion zone under a constant heat ratio of arc to laser. Heat accumulation in a narrow area and the undercooling degree are the two main factors for the formation of columnar crystal. The ME21/ME21 joint has a tensile strength of up to 185.2 MPa, which is about 81.8% of that of the ME21 base metal (BM-ME21). The tensile strength of the ME21/AZ31 joint (158.8 MPa) is lower than that of the ME21/ME21 joint. The fracture of the ME21/ME21 and ME21/AZ31 joints occurs at the junction of the columnar crystal and the heat-affected zone. The microhardness of the ME21/AZ31 joint presents a low–high–low–high trend from BE-ME21 to BE-AZ31, and the distribution of the ME21/AZ31 welded joint microhardness in the cross-section presents a low–high–low trend. The ME21/ME21 weld seam is composed of an AlCe_3_ intermetallic compound, Mn particles, and α-Mg, and the ME21/AZ31 weld seam is composed of a α-Mg, Mg_17_Al_12_, and AlCe_3_ intermetallic compound.

## 1. Introduction

In order to meet the demands of automobiles for materials of light weight and low energy consumption, developing the Mg alloy and Al alloy plate for automotives to replace the widely used steel plate is an effective method at present [1,2,3,4,5]. The Mg alloys used in automobiles are mainly Mg–Al–Zn alloys [6], Mg–Zn–Zr alloys [7], Mg–A1–Re alloys [8], etc. [9].

The single kind of Mg alloy has been unable to meet the performance requirements of different automotive parts. Therefore, the same car may use a variety of Mg alloys [10]. At present, the research on Mg alloy sheet (around 1.5-mm thickness) welding mainly focuses on the same Mg alloy [11,12]. However, there are a few research studies on the welding of dissimilar Mg alloy sheets for automobiles, and the study is not systematic and comprehensive enough. ME21 is a recently developed Mg–Mn–Ce alloy that is characterized by high corrosion resistance and good ductility, which has the potential for wide use in the automobile industry. However, the research on the ME21 Mg alloy welding is less involved.

At the moment, many scholars have conducted extensive research on the welding of similar and dissimilar Mg alloys [13,14,15]. There usually are major methods for Mg alloy welding: laser welding, Tungsten Inert Gas (TIG) welding, Metal Inert Gas (MIG) welding, Friction Stir Welding (FSW), hybrid heating source welding, etc. Bailey et al. [14] studied parameterization on laser welding of 2.0-mm thick AZ31 plates by a fiber laser. They found that the width, grain size, and secondary dendrite arm spacing were all affected more strongly by welding speed than by laser power for butt welding. Coelho et al. [16] investigated the microstructure–property relation of an AZ31B laser beam weld without filler wire. They concluded that the heat-affected zone is very narrow, and the microhardness values are almost uniform across the base material of the heat-affected zone (HAZ) and fusion zone. Carlone et al. [13,17] welded 4-mm thick Mg alloy plates with TIG welding and FSW welding, and then compared the joints with different welding methods. They found that very small globular grains formed in the fusion zone for TIG joints, and the refined grains that were attributable to partial recrystallization phenomena were individuated in both the thermo-mechanical affected zone and the heat-affected zone.

The traditional TIG welding quality is good; it has a wide range of application, and is suitable for all kinds of welding joints [18,19]. However, the energy density of TIG welding is small, and the HAZ is wide. Laser welding HAZ is narrow, and forms a good-looking welding seam [20,21,22]. Nevertheless, it is easy to generate gas holes in the welding seam, and the cost of laser welding is high [23]. The laser–TIG hybrid heating source welding not only gives full play to their respective advantages, but also makes up for each other’s deficiencies. Laser–TIG welding is a high-efficiency welding method with the advantages of low cost, good welding quality, and stable welding, which can meet the requirements of efficient production in the automotive industry [24,25,26].

The effects of rare earth elements on the microstructures and properties of Mg alloys have been studied extensively at present. However, few research studies on the welding of rare earth Mg alloy sheets (1.4-mm thickness) for automobiles have been carried out. This paper discusses some experimental results concerning the microstructural and mechanical properties of ME21/ME21 and ME21/AZ31 butt joints welded through laser-TIG, and compares them with AZ31/AZ31 butt joints to find some rules regarding the welding process.

## 2. Experimental Procedures

The AZ31 Mg alloy and ME21 Mg alloy were processed by hot rolling. Rolling sheets of two commercial Mg alloys AZ31 (2.86 Al, 0.83 Zn in wt.%, Mg balance) and ME21 (2.34 Mn, 0.55 Ce in wt.%, Mg balance) with thickness of 1.4 mm were used for the present work. The Mg-based filler wire (AZ61) with the diameter of 1.6 mm was utilized.

A pulsed Yttrium Aluminum Garnet (YAG) laser with a maximum power of 1 kW and a TIG welding machine were applied. The minimum diameter of the laser spot was 0.6 mm, and the laser wavelength was 1064 nm. The axis of the laser beam was put perpendicularly to the workpiece, and the angle between the TIG torch and the beam was 45°. The welding parameters were fixed as follows: the laser power was 500 W, the wire feeding speed was 2100 mm/min, the welding speed was 1900 mm/min, the argon flow of the welding area’s top surface and bottom surface was 10 L/min, the AC–TIG welding current was 80 A, the welding voltage was 11 V, and the diameter of the pure tungsten electrode was 2.4 mm. The formula for the laser–TIG hybrid heat source (Q_hybrid_) was as follows: Q_hybrid_ = (P_laser_ + I_arc_·U_arc_)·η/V_speed_. P_laser_ was laser power, 500 W. I_arc_ and U_arc_ were the welding current and voltage, 80 A and 11 V, respectively. η was the welding efficiency, 0.7 [27]. V_speed_ was the welding speed, 1900 mm/min. Hence, the welding heat input was 30.5 J/mm.

The distance between the head of the TIG torch and the workpiece surface was 1.5 mm, and the distance between the TIG torch head and the axis of the laser beam was 1 mm. A schematic diagram of the Mg alloy welding process is shown in Figure 1.

The welds were etched by solution containing 5 mL of nitric acid, 5 mL of oxalic acid, 90 mL of water, and the etching time was about 15 s to reveal the microstructure. The microstructure and fracture surface were examined by Leica-MEF4 optical microscope and SEM (scanning electron microscope), respectively. An electron probe micro-analyzer (EPMA) was used to detect the element distributions in the weld seam. The phase compositions of weld and base metal were identified by a PANayltical (Almelo, Netherlands) Empyrean powder X-ray diffractometer. The Vickers microhardness maps of typical welds were measured on polished surfaces across the weld. The tensile sample size was shown in Figure 9, and the stretching speed was 0.5 mm/min at room temperature. All the specimens were moved to the weld reinforcement before the tensile testing. In order to observe the weak position of joints and calculate the tensile strength, the weld reinforcements were removed before the tensile testing.

## 3. Results and Discussion

### 3.1. Bead Shape

As shown in Figure 2, the welds exhibited uniform and beautiful appearances in the same condition. The morphology of the top and bottom surface of the ME21/ME21, AZ31/AZ31, and ME21/ME21 joints were different. The ME21/ME21 welding width of the bottom surface was significantly smaller than that of the AZ31/AZ31 joints. Studies have shown that the thermal conductivity of the cast samples for Mg–0.5Mn–0.15Ce and Mg–0.5Mn–0.6Ce at room temperature were 138.2 W/(m·K) and 129.9 W/(m·K), respectively, and the thermal conductivity of the AZ31 alloy is 76.9 W/(m·K) [28,29]. Wang et al. pointed out that the thermal conductivity of the Mg–(1.5–2.5)Mn–(0.15–0.35)Ce alloy is 134 W/(m·K) at 100 °C [30]. The signs suggest that the thermal conductivity of ME21 alloy is higher than that of the AZ31 alloy. Compared with the AZ31 alloy, the larger the thermal conductivity of the ME21 alloy, the more narrow the welding seam width that will be gained. It could be seen that the stability of the weld shape of same metal is superior to that of a dissimilar metal.

### 3.2. Microstructure Characteristics

As shown in Figure 3, the microstructures of the base metal AZ31 (BM-AZ31) were composed of massive equiaxed grains with non-uniform sizes. The average grain size of the BM-AZ31 was observed to be 5–10 μm. The base metal ME21 (BM-ME21) structure is homogeneous, and the average grain size crystal diameter is about 25 μm.

Low magnification micrographs of the cross-sections of the ME21/ME21, AZ31/AZ31, and ME21/AZ31 joints are shown in Figure 4. The boundaries between the base metal and weld seam are clearly seen in the two kinds of joints. Due to the chemical etching, the weld seam is dark, which is distinct from the base metal in the micrographs. It could be seen that the angle of the AZ31/AZ31 joint (64.77°) between the fusion line and the lower surface of the plate is larger than that of the ME21/ME21 joint (50.62°). The main reason for these differences is that the thermal conductivity of ME21 is larger than that of AZ31. During the welding process of ME21 and AZ31, more heat is transferred to the ME21 side, and the angle of both sides are reduced compared to the ME21/ME21 joint and AZ31/AZ31 joint, as shown in Figure 4c. It could be seen that both the welding of dissimilar metal materials and the same metal materials have obvious influences on the weld section.

Figure 5 shows the microstructure of the ME21/ME21 joint, the AZ31/AZ31 joint, and the ME21/AZ31 joint, whose selected positions were shown in Figure 4. As shown in Figure 5a, the fusion line of the upper part joint (I) of ME21/ME21 is clearly seen at the border of HAZ (HAZ) and the fusion zone (FZ). Figure 5b shows the microstructure of the lower part of the joint (II) of ME21/ME21. The width of HAZ on zone II of ME21/ME21 is larger than that on zone I. It also can be seen that columnar crystals appeared between HAZ and FZ, which grew along the direction perpendicular to the fusion line on the lower part of the joint. Fine crystals also were found between the columnar crystals zone (CCZ) and HAZ. The above phenomenon also appears in the ME21/AZ31 joint, which is close to BM-ME21, as shown in Figure 5f. However, columnar crystals are not found in another side of the ME21/AZ31 joint, which is close to BM-AZ31, as shown in Figure 5c,d. No columnar crystal was formed during welding of the AZ31 Mg alloy [31]. On the one hand, this is mainly because the heat accumulated in narrow areas of the lower part of the joint due to the combined heat source effect of laser and arc welding. On the other hand, due to the thermal conductivity of ME21 being larger than that of AZ31, the undercooling degree of the fusion line that is close to BM-ME21 is relatively large [28,29,30]. So, it is easy to form coarse columnar crystals along the direction perpendicular to the fusion line of the ME21 side instead of the AZ31 side.

EPMA was used to observe the microstructure of the weld seam and columnar crystal, and Figure 6 shows the microstructures of the ME21/AZ31 joint. Coarse columnar crystals are clearly visible, and many white bright phases are found to precipitate along the grain boundaries in the CCZ. Many fine particles precipitate along the grain boundaries in the FZ as well. The precipitate phases distribute in CCZ continuously and distribute in FZ dispersively. The results of EPMA-Energy Dispersion Spectrum (EDS) analysis of the microstructure are shown in Table 1. EPMA–EDS analysis indicated that P1 and P2 in the CCZ contain Mg_95_Ce_3_Mn_1_Al_1_ (at.%) and Mg_99_Mn_1_ (at.%). The Ce element mainly precipitates along the grain boundaries and the Mn element is distributed along the grain boundaries or in the grain interiors. P3 analysis results are Mg_55_Ce_3_Mn_11_Al_31_ (at.%), P4 analysis results are Mg_97_Al_3_ (at.%). The welds mainly consist of α-Mg and Mg_17_Al_12_ in the AZ31/AZ31 joint. In addition to Mg and Al elements, a large number of Mn and Ce elements precipitate at the grain boundaries in the FZ.

Figure 7 shows the secondary electron image and elements distribution by EPMA. It can be seen that the distribution of Mg and Mn elements is more uniform in the whole scanning area. However, there are two extremes in the distribution of the Al element. The Al elements are almost absent in the HAZ and CCZ, and there are more Al elements in the FZ. No Al elements existed in the ME21 Mg alloy, but they did appeared in the FZ. It can be observed that the grain size in CCZ was larger than that in BM-ME21, and few Al elements existed in CCZ. Therefore, the HAZ contains the CCZ.

Heat accumulation in a narrow area, high thermal conductivity, and undercooling degree are the main factors for the formation of columnar crystal. Under a large undercooling degree, the liquid metal of FZ first solidified near the fusion line. At the same time, the residual heat moves along the direction perpendicular to the fusion line to the BM-ME21. Due to the heat accumulation in a narrow area and high undercooling degree, the grain of HAZ in the lower part of the weld will grow along perpendicular to the direction of the fusion line to form the columnar crystal. With the liquid metal of FZ solidification, the columnar crystals will no longer grow. The value of the heat input directly affects the shape of the columnar crystals [32]. Figure 5b also found that a fine grain zone formed along the columnar crystal region in the HAZ. During the cooling process, recovery and recrystallization occur in part of the grains that are near the CCZ, resulting in grain refinement [33].

Figure 8 shows the X-ray diffraction (XRD) spectra of ME21 and a three-joint weld seam. The locations of the weld seam XRD phase analysis are shown in Figure 8a, which is located in the center of the weld seam. Figure 8b shows the phase compositions of the ME21 base metal. It can be seen that the ME21 Mg alloy is mainly composed of Mg_12_Ce, Mn particles, and α-Mg [28,34]. As shown in Figure 6, lots of precipitate phases formed in the weld seam. During the laser–TIG welding of ME21 with AZ61 filler wire, an AlCe_3_ intermetallic compound is detected in the weld seam, and there were no Mg_12_Ce forms based on the analysis of XRD, as shown in Figure 8c. The ME21/ME21 weld seam is composed of an AlCe_3_ intermetallic compound, Mn particles, and α-Mg. Figure 8d shows the XRD analysis of the AZ31/AZ31 weld seam; the weld seam is mainly composed of an α-Mg and Mg_17_Al_12_ intermetallic compound. Figure 8e shows the XRD analysis of the ME21/AZ31 weld seam. The phase compositions of the ME21/AZ31 weld seam are different from ME21/ME21 and AZ31/AZ31, whose compositions consist of a α-Mg, Mg_17_Al_12_, and AlCe_3_ intermetallic compound.

### 3.3. Mechanical Properties

The tensile strength of specimens is tested, and the result is shown in Figure 9. The test results are the average of three samples. The tensile strength of BM-ME21 is 226.3 MPa, while that of the ME21 butt plate is reduced to 181.3 MPa, which is about 80.1% of that of BM-ME21. At the same experimental condition, the tensile strength of the AZ31 butt plate reached 226.1 MPa, and obtained nearly 82.7% compared with BM-AZ31 (273.3 MPa). However, the tensile strength of the ME21/AZ31 butt plate (158.7 MPa) is not between the ME21 and AZ31 butt plates, which is lower than the tensile strength of the ME21 butt plate.

Figure 10 shows the fracture location of Mg alloys joints. As shown in Figure 10a, the ME21/ME21 joint mostly fractured at FZ, and the others fractured at CCZ, which belongs to the bottom of the weld seam. It can be seen in the magnification of CCZ that the other microstructure in the HAZ is different from that of the columnar crystal; it is easy to break at the junction of HAZ and CCZ. As shown in Figure 10b, the AZ31/AZ31 joints mostly fractured at the FZ, and partial cracks propagated along the fusion line of LPJ, which is different from the fracture mechanism of the ME21/ME21 joint. The ME21/AZ31 joint fractured at FZ and CCZ, which is similar to the fracture mechanism of the ME21/ME21 joint, as shown in Figure 10c. However, the tensile strength of the ME21/AZ31 joint is lower than that of the AZ31/AZ31 and ME21/ME21 joints, which is mainly because the Ce and Mn content in the ME21/AZ31 joint is lower than that in the ME21/ME21 joint, which is confirmed to be conducive to improving the strength of joint. The results indicate that the joint quality of dissimilar Mg alloy welding of ME21 and AZ31 is significantly lower than that of the same Mg alloy welding. Columnar crystal Ce and Mn contents are the main reason for the low tensile properties of the ME21/AZ31 joint.

Due to the big differences in performance between the columnar grain and fine-grained properties, the performance of columnar crystal was relatively poor in the direction perpendicular to the columnar crystal. The crack first grows along the juncture of the CCZ and HAZ, which was perpendicular to the columnar crystal in the stretch process of the ME21/ME21 joint and the ME21/AZ31 joint. Then, it went through the columnar crystal and grew parallel to it, and finally extended toward the FZ.

As shown in Figure 11a, the fracture surface of the ME21/ME21 joint is characterized by some quasi-cleavage fracture with remarkably tearing ridges, but also shows some brittle features. The Figure 11b fracture surfaces’ appearance is different from Figure 11a, an SEM fractograph revealed that intergranular cracking, transgranular cracking, and cleavage steps were present on the tensile fracture. The ME21/ME21 welding fracture is in primarily brittle features.

The fracture surfaces of the AZ31/AZ31 joints are shown in Figure 11c,d. The fracture morphology such as cleavage steps and dimples exist in Figure 11c, while porosity is observed on that of Figure 11d. The AZ31/AZ31 welding fracture acted as a hybrid fracture mod. The ME21/AZ31 fracture in Figure 11e shows signs of a distinctly different feature to that in Figure 11f. The fracture surface of Figure 11e exhibits a tearing ridge, which possesses a definite directional character. However, the fracture surface of Figure 11f exhibits small dimples, which indicates that the seam has better toughness.

The microhardness of the ME21/ME21 and ME21/AZ31 weld metal zone in the cross-section are both tested. The tests points of ME21/ME21 are distant to the upper surface of the weld—0.3 mm, 0.6 mm, 0.9 mm, and 1.2 mm on the vertical axis, and take a point on the horizontal every 0.5 mm. The ME21/AZ31 microhardness test points are 0.2 mm, 0.5 mm, 0.8 mm, and 1.1 mm from the upper surface. The hardness of the butt joints was measured using a Vickers microhardness tester with a 100-gram load for a 15-second dwell time. The microhardness of the AZ31 and ME21 base metal were measured to be 49.6 HV_0.1_ and 71.0 HV_0.1_ on average, respectively.

The hardness measurement values are shown in Figure 12. The microhardness of the ME21/ME21 weld seam is higher than that of ME21. Moreover, the distribution of microhardness in the cross-section presents a low–high–low trend, and the lowest microhardness is that of the BM-ME21. However, the microhardness of the ME21/AZ31 weld seam presented a low–high–low–high trend from the ME21 base metal to BM-AZ31, and the highest microhardness is in the fusion zone, which is close to the side of BM-ME21. There is a region of low microhardness between the weld center and BM-AZ31. Ce and Mn can inhibit the growth of the AZ61 alloy grain to improve the mechanical properties [28]. During the welding process, the Ce and Mn elements melted into the weld, and played a role in refining the AZ61 grain, so the microhardness of the weld seam improved remarkably. This is a reason why the microhardness close to the side of the ME21 alloys is higher than that in the other region.

## 4. Conclusions

AZ31 and ME21, AZ31 and AZ31, and ME21 and ME21 Mg alloy sheets were successfully welded by laser–TIG hybrid welding with filler wire. The laser power, arc current, and welding speed were 500 W, 80 A, and 1900 mm/min, respectively. The welding heat input was 30.5 J/mm.Columnar crystals appeared in the ME21/ME21 and ME21/AZ31 FZ, which grew along the side wall of the HAZ, and no columnar crystals formed in the AZ31/AZ31 fusion zone at the same experimental conditions. The HAZ contains the CCZ.The ME21/ME21 weld seam is composed of an AlCe_3_ intermetallic compound, Mn particles, and α-Mg, and the ME21/AZ31 weld seam is composed of a α-Mg, Mg_17_Al_12_, and AlCe_3_ intermetallic compound.The ME21/ME21 and ME21/AZ31 joints are broken at the junction of the columnar crystal and the heat-affected zone. The ME21/ME21 joint has a tensile strength of up to 185.2 MPa, which is about 81.8% of BM-ME21. The ME21/AZ31 joint (158.8 MPa) is lower than the tensile strength of the ME21/ME21 joint.

## Figures and Tables

**Figure 1 materials-12-02188-f001:**
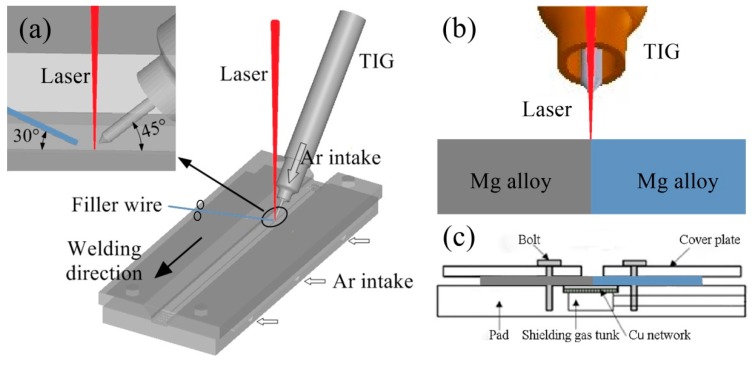
(**a**) Schematic diagram of Mg alloy welding process; (**b**,**c**) the cross-section of Mg alloy plates and welding fixture.

**Figure 2 materials-12-02188-f002:**
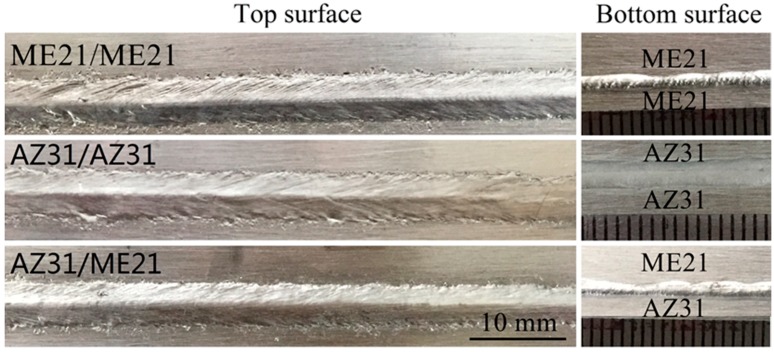
Bead surface appearances of welded joints.

**Figure 3 materials-12-02188-f003:**
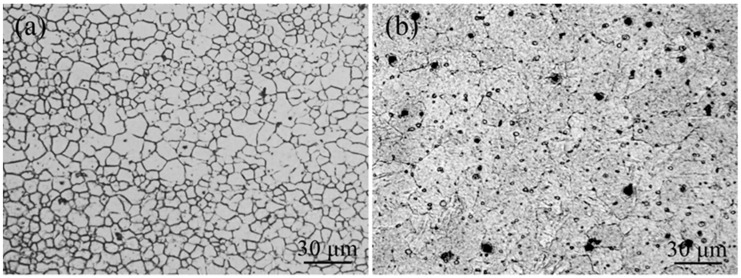
Microstructure of the base metal (**a**) AZ31 Mg alloy; (**b**) ME21 Mg alloy.

**Figure 4 materials-12-02188-f004:**
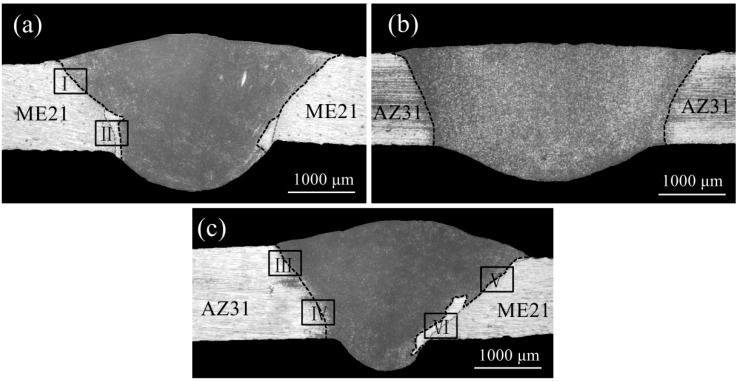
Micrographs of the joint cross-sections (**a**) ME21/ME21 joint; (**b**) AZ31/AZ31 joint; (**c**) ME21/AZ31 joint.

**Figure 5 materials-12-02188-f005:**
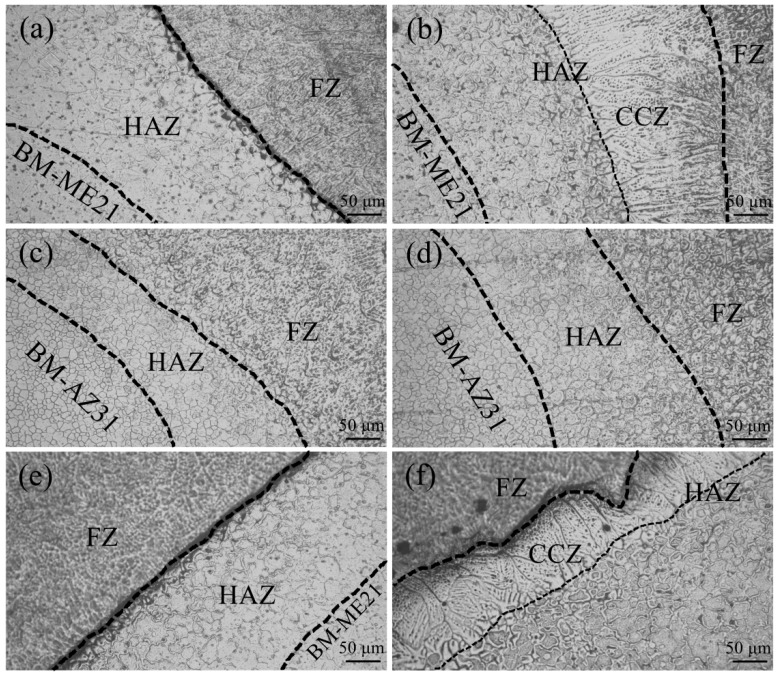
Microstructures of different joints (**a**) zone I in Figure 4a; (**b**) zone II in Figure 4a; (**c**) zone III in Figure 5c; (**d**) zone IV in Figure 4c; (**e**) zone V in Figure 4c; (**f**) zone VI in Figure 4c.

**Figure 6 materials-12-02188-f006:**
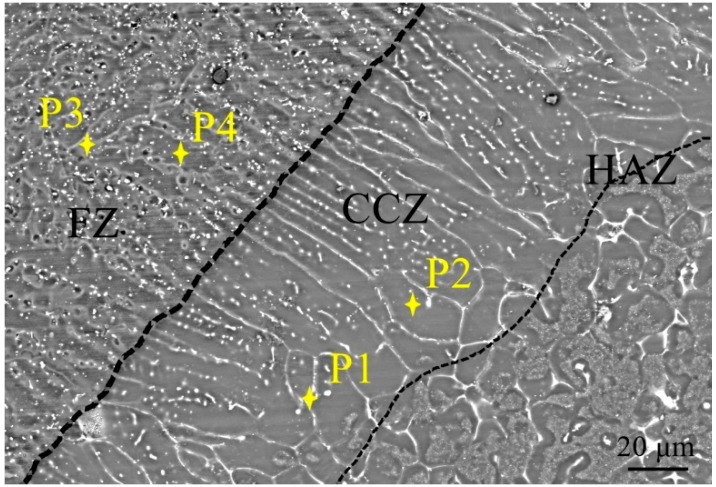
Microstructures of ME21/AZ31 joint by electron probe micro-analyzer (EPMA).

**Figure 7 materials-12-02188-f007:**
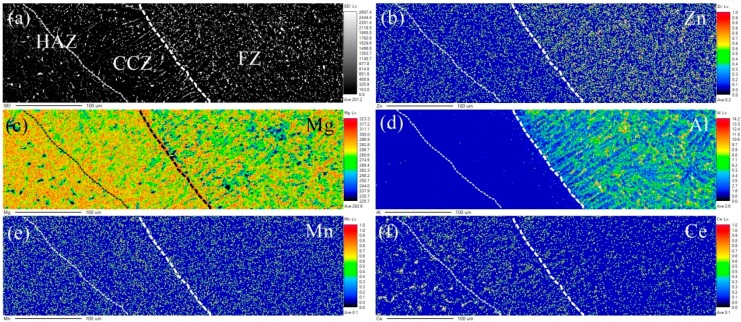
(**a**) Secondary electron image by EPMA; (**b**–**f**) elements distribution of Zn, Mg, Al, Mn, and Ce in (**a**).

**Figure 8 materials-12-02188-f008:**
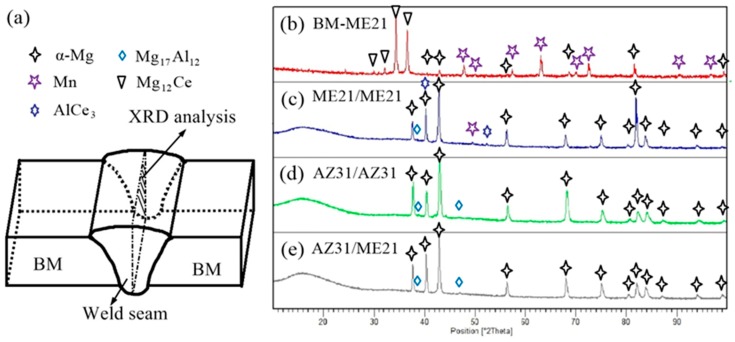
XRD spectra of ME21 and weld seam: (**a**) XRD phase analysis location; (**b**) ME21 base metal; (**c**) ME21/ME21; (**d**) AZ31/AZ31; and (**e**) ME21/AZ31.

**Figure 9 materials-12-02188-f009:**
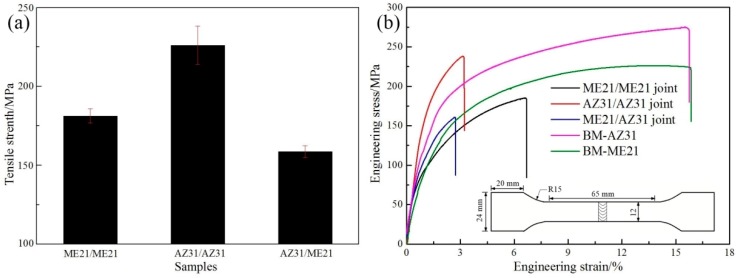
(**a**) Tensile strength of different joints; (**b**) Engineering stress–strain curve of different joints.

**Figure 10 materials-12-02188-f010:**
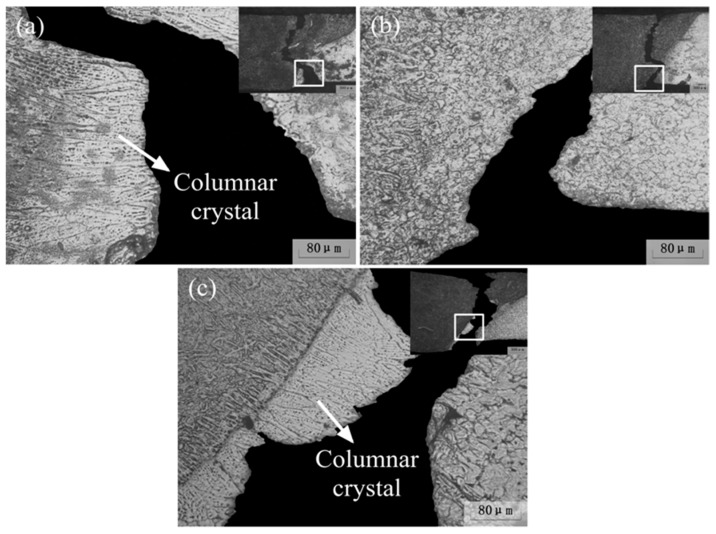
The fracture location of Mg alloys joints: (**a**) ME21/ME21; (**b**) AZ31/AZ31; (**c**) ME21/AZ31.

**Figure 11 materials-12-02188-f011:**
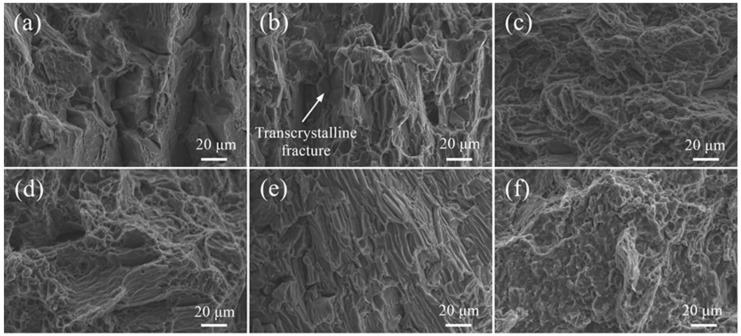
SEM morphologies of fracture surfaces at high magnifications: (**a**) ME21/ME21 at the top half of the weld; (**b**) ME21/ME21 at the bottom half of the weld; (**c**) AZ31/AZ31 at the top half of the weld; (**d**) AZ31/AZ31 at the bottom half of the weld; (**e**) ME21/AZ31 at the top half of the weld; (**f**) ME21/AZ31 at the bottom half of the weld.

**Figure 12 materials-12-02188-f012:**
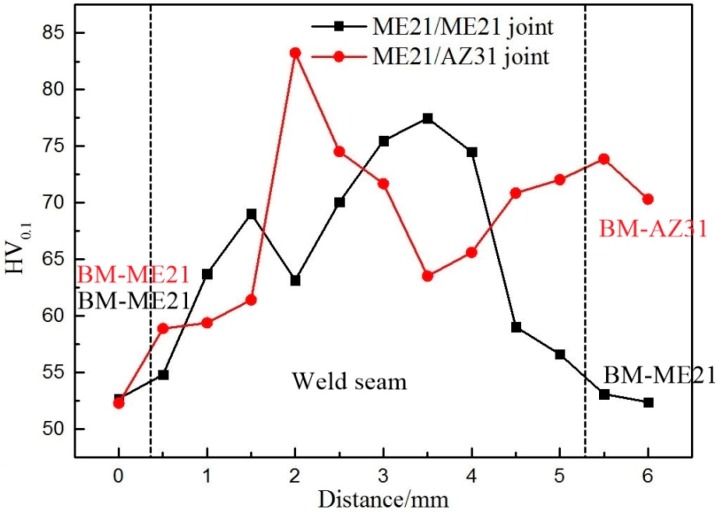
Microhardness map of ME21/ME21 joint and ME21/AZ31 joint.

**Table 1 materials-12-02188-t001:** EPMA-EDS analysis of the ME21/AZ31 joint in Figure 6.

Point	Atomic Content (%)				
	Mn	Ce	Al	Mg	Possible Phase
P1	0.53	3.42	0.58	95.47	Mg_95_Ce_3_Mn_1_Al_1_
P2	0.83	0.03	0.03	99.11	Mg_99_Mn_1_
P3	11.03	3.24	30.87	54.86	Mg_55_Ce_3_Mn_11_Al_31_
P4	0.02	0.0008	3.07	96.91	Mg_97_Al_3_

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
