# Peer review of "Mechanical Properties and Microstructures of Laser–TIG Welded ME21 Rare Earth Mg Alloy"

_materials, 2019, doi:10.3390/ma12132188_

Reviewer 1 Report

Although this manuscript is well organized and the methods are suitable, I do not find any scientific novelty in this paper.

In some sentences there is a lack of verbs, for example "However, very few research on welding of rare earth magnesium alloy sheet (1.4 mm thickness) for automobile."

etc.

"tensile strenth" ?

etc.

Some sentences are grammatically incomprehensible, for example:

"The tensile samples shown in Fig.9 and a constant crosshead speed of 0.5 mm/min at room temperature, and the test results are the average of three samples."

"The fracture surfaces of AZ31/AZ31 welding [...]" ?

etc.

The paper is not acceptable for publication in its present form. Throughout the paper the language needs significant improvement.

Reviewer 2 Report

Line 67: Please describe weld bevel geometry of the base plates!

Line 72: In order to enable repeatibility of the experiment please define type of current for TIG (DC or AC), type and diameter of tungsten  electrode as well as welding voltage .

Please calculate and insert welding energy (heat input) for TIG!

Conclusion  remarks:Please refer towards applied energy of the TIG process and laser process. This results should be connected with this specific energy input and not be taken as a general consclusion for all welding energy levels.

Reviewer 3 Report

Interesting works. There are some points that needs to be revised as follows.

The English needs to be revised by a native speaker. There are several mistakes in the manuscript.

The introduction should be updated with some key works. For example it is important to put into context the topic on laser welding of Mg alloys (see A review of laser welding techniques for magnesium alloys), provide the advantages of TIG (see Effect of TIG current on microstructural and mechanical properties of 6061-T6 aluminium alloy joints by TIG–CMT hybrid welding and Tungsten inert gas (TIG) welding of Ni-rich NiTi plates: functional behavior) and laser (Laser welding of precipitation strengthened Ni-rich NiTiHf high temperature shape memory alloys: Microstructure and mechanical properties and Liquid metal embrittlement in laser beam welding of Zn-coated 22MnB5 steel) welding of advanced engineering alloys. Please update the references accordingly.

What was the beam spot size, wavelengthsand selected power? The welding speed is not provided as well.

“All specimens were moved the weld reinforcement before the tensile testing”: what does this mean?

In welding there are no CCZ region. This is clearly part of the HAZ. Please revise.

Can EDS in a SEM actually quantify 1% of Mn and/or Al? SEM is a semi-quantitative technique.

Please also provide the tensile curve of the base materials.

In figure 12 please use the correct aspect ratio in origin to show the hardness maps. Clearly height of the graphs should be reduced. See an example of this in Microstructure and mechanical properties of gas tungsten arc welded Cu-Al-Mn shape memory alloy rods.

Author Response

Round  2

Reviewer 3 Report

The authors improved the manuscript after revision